# AVERAGE CONTROLLED AND AVERAGE NATURAL MICRO DIRECT EFFECTS IN SUMMARY CAUSAL GRAPHS

## ABSTRACT

In this paper, we investigate the identifiability of average controlled direct effects and average natural direct effects in causal systems represented by summary causal graphs, which are abstractions of full causal graphs, often used in dynamic systems where cycles and omitted temporal information complicate causal inference. Unlike in the traditional linear setting, where direct effects are typically easier to identify and estimate, non-parametric direct effects, which are crucial for handling real-world complexities, particularly in epidemiological contexts where relationships between variables (*e.g.*, genetic, environmental, and behavioral factors) are often non-linear, are much harder to define and identify. In particular, we give sufficient conditions for identifying average controlled micro direct effect and average natural micro direct effect from summary causal graphs in the presence of hidden confounding. Furthermore, we show that the conditions given for the average controlled micro direct effect become also necessary in the setting where there is no hidden confounding and where we are only interested in identifiability by adjustment.

## 1 INTRODUCTION

The identification and estimation of direct effects are critical in many applications. For instance, epidemiologists aim to measure how smoking affects lung cancer risk independently of genetic susceptibility (Zhou et al., 2021). In addition, it has been shown that estimating direct effects can help diagnose system failures by comparing the direct impacts of different components before and after failure (Assaad et al., 2023).

In the framework of Structural Causal Models (SCMs) (Pearl, 2009), there has been significant progress in identifying direct effects using fully specified causal graphs. However, constructing a fully specified causal graph is a challenging task, as it requires precise knowledge of the causal relationships between all pairs of observed variables. In complex, high-dimensional settings such as in medico-administrative databases, this level of detail is often unavailable, limiting the practical application of causal inference. As a result, there has been growing interest in the use of partially specified causal graphs in recent years (Perkovic, 2020; Anand et al., 2023; Ferreira & Assaad, 2024a; Assaad et al., 2024; Wahl et al., 2024). One notable example is the cluster-ADMG (Acyclic Directed Mixed Graph) introduced by Anand et al. (2023), which offers a coarser representation of causal relationships. In this graph, vertices represent clusters of variables and therefore it simplifies the representation of complex systems. However, one main limitation of this type of graph is its acyclicity assumption which does not necessarily hold even if the underlying fully specified graph is acyclic. A closely related graph type is the summary causal graph (SCG) (Peters et al., 2013), which is mainly used in systems involving time and which relaxes the assumption of acyclicity and represents causal relationships where each cluster corresponds to a time series or repeated measurements of the same variable in longitudinal studies. This flexibility makes SCGs particularly useful in scenarios involving temporal data and cycles.

In SCGs, two types of causal effects may be of interest. The first is the macro causal effect, which captures the impact of one set of time series on another set of time series. The second is the micro causal effect, which concerns the effect of one specific time point on another. This paper focuses on micro causal effects, with particular emphasis on micro direct effects. To date, the problem of identifying micro direct effects from SCGs has primarily been addressed under the assumptions of

no hidden confounding and linear SCMs (Ferreira & Assaad, 2024a). However, many real-world applications involve hidden confounding and relationships between causes and effects that are not necessarily linear. In a non-parametric setting, defining direct effects becomes more complex. The literature of causal inference primarily offers two definitions: the controlled direct effect and the natural direct effect (Robins & Greenland, 1992; Pearl, 2001). The former isolates the impact of the treatment variable by controlling on mediators or on the parents of the response variable, while the natural direct effect captures how the treatment variable influences the response variable while fixing mediators to their natural values, making it more realistic but harder to identify and estimate.

This paper focuses on the identifiability of direct effects from SCGs in a non-parametric setting. Our contributions are threefold: first, we present sufficient conditions that characterizes cases where the average controlled micro direct effect is graphically identifiable from an SCG in the presence of hidden confounding. Next we show that those conditions become necessary in the absence of hidden confounding and when considering only identifiability by adjustment. Finally, we establish sufficient conditions for the identifiability of the average natural micro direct effect from an SCG with hidden confounding.

The remainder of the paper is organized as follows: Section 2 reviews related work, while Section 3 introduces preliminaries and tools necessary for the subsequent sections. Section 4 provides identifiability conditions for average controlled micro direct effects, and Section 5 addresses identifiability conditions for average natural micro direct effects. Finally, Section 7 concludes the paper.

## 2 RELATED WORKS

There is a substantial body of work in the literature focused on identifying causal effects using partially specified graphs. For instance, the studies by Maathuis & Colombo (2013); Perkovic et al. (2016); Perkovic (2020); Wang et al. (2023) provided conditions for identifying total effects through partially directed graphs that represent all graphs within the Markov equivalence class of the true causal graph, including CPDAGs, MPDAGs, and PAGs. Furthermore, Flanagan (2020) provided conditions for identifying average controlled direct effects and average natural direct effects using CPDAGs and MPDAGs assuming no hidden confounding.

In the context of Cluster-ADMGs (acyclic directed mixed graphs where each vertex represents a group of variables), Anand et al. (2023) extended Pearl's do-calculus (Pearl, 1995) to establish necessary and sufficient conditions for identifying total effects between clusters. Building on this, Ferreira & Assaad (2024b) extended these results to SCGs, where each cluster represents a time series and which unlike Cluster-ADMGs, can contain cycles. However, both studies focus on macro causal effects (*i.e.*, effects between clusters), whereas the present paper is concerned with micro-level causal effects within clusters.

For micro causal effects, Eichler & Didelez (2007) provided sufficient conditions for identifying micro total effects in time series, applicable to SCGs under the assumption of no instantaneous relations. When instantaneous relations are allowed, Assaad et al. (2023) showed that both micro total and micro direct effects are identifiable from SCGs in a linear setting, assuming no hidden confounders and that cycles in the SCG cannot be of size greater than one. Subsequently, Ferreira & Assaad (2024a) and Assaad et al. (2024) extended these results, establishing conditions for identifying micro total effects in a non-parametric setting and micro direct effects in a linear setting, even when cycles larger than one exist in the SCG. More recently, Assaad provided sufficient conditions for identifying total effects in the presence of hidden confounding. However, identifying micro direct effects in a non-parametric setting or with hidden confounding from SCGs (or any graph with cycles) remains an open challenge. This is not surprising, as direct effects are inherently more complex than total effects in non-parametric setting.

## 3 PRELIMINARIES

In this section, we introduce the tools, terminology, and key theoretical results that will be essential for the remainder of the paper.

In the remainder, uppercase letters are used to denote variables, lowercase letters represent specific values of those variables, and blackboard bold letters indicate sets of variables. For any given graph,

$Parents(\cdot,\cdot)$, $Ancestors(\cdot,\cdot)$, and $Descendants(\cdot,\cdot)$ will be used to represent the parents, ancestors, and descendants of a vertex, respectively. The first argument refers to the specific vertex under consideration, while the second argument corresponds to the graph in which these relationships are being examined. For instance, $Parents(X, \mathcal{G})$ refers to the parents of vertex $X$ in graph $\mathcal{G}$. By convention, we will treat any vertex as an ancestor and a descendant of itself. The mutilated graph $\mathcal{G}_{\overline{W}\underline{Z}}$ is obtained from $\mathcal{G}$ by removing all edges with an arrowhead pointing to $W$ (e.g., $X \to W$, $X \leftarrow\!\dashrightarrow W$), as well as all edges with a tail originating from $Z$ (e.g., $X \leftarrow Z$). In a graph, a path is said to be activated by $\mathbb{W}$ if every collider $W$ (i.e., a vertex with structure $\to W \leftarrow$) on the path, or any of its descendants, is in $\mathbb{W}$, and no non-collider on the path is in $\mathbb{W}$. A path that is not activated is said to be blocked. If all paths between $X$ and $Y$ are blocked by a set $\mathbb{W}$, then $X$ is said to be d-separated from $Y$ by $\mathbb{W}$ in the graph $\mathcal{G}$, denoted as $X \perp\!\!\!\perp_{\mathcal{G}} Y \mid \mathbb{W}$. The strongly connected component of a vertex $X$ in a graph $\mathcal{G}$, denoted by $Scc(X, \mathcal{G})$, is the maximal subset of vertices that includes $X$, such that every vertex in this subset has a directed path to every other vertex in the subset. By convention, $Scc(X, \mathcal{G}) = \varnothing$ if there is no cycle in the graph that includes $X$ and $Scc(X, \mathcal{G}) = \{X\}$ if the only cycle that includes $X$ is a self loop on $X$. We refer to $\Pr$ as the probability distribution, and $\Pr(\cdot \mid do(X = x))$ as the interventional distribution, where the $do(\cdot)$ operator represents the intervention.

In this study, we consider an unknown discrete-time dynamic structural causal model, a specific variant of the structural causal model (Pearl, 2009), designed to account for temporal dynamics.

**Definition 1** (Discrete-time dynamic structural causal model (DTDSCM)). *A discrete-time dynamic structural causal model is a tuple $\mathcal{M} = (\mathbb{L}, \mathbb{V}, \mathbb{F}, \Pr(\mathbb{l}))$, where $\mathbb{L} = \bigcup\{\mathbb{L}^{v_t^i} \mid i \in [1, d], t \in [t_0, t_{max}]\}$ is a set of exogenous variables, which cannot be observed but affect the rest of the model. $\mathbb{V} = \bigcup\{\mathbb{V}^i \mid i \in [1, d]\}$ such that $\forall i \in [1, d]$, $\mathbb{V}^i = \{V_t^i \mid t \in [t_0, t_{max}]\}$, is a set of endogenous variables, which are observed and every $V_t^i \in \mathbb{V}$ is functionally dependent on some subset of $\mathbb{L}^{v_t^i} \cup \mathbb{V}_{\leq t} \backslash \{V_t^i\}$ where $\mathbb{V}_{\leq t} = \{V_{t'}^j \mid j \in [1, d], t' \leq t\}$. $\mathbb{F}$ is a set of functions such that for all $V_t^i \in \mathbb{V}$, $f^{v_t^i}$ is a mapping from $\mathbb{L}^{v_t^i}$ and a subset of $\mathbb{V}_{\leq t} \backslash \{V_t^i\}$ to $V_t^i$. $\Pr(\mathbb{l})$ is a joint probability distribution over $\mathbb{L}$.*

Some of the findings in this paper consider an unknown, general DTDSCM, while others focus on a specific subfamily of DTDSCM that are constrained by the following assumption.

**Assumption 1.** $\forall L, L' \in \mathbb{L}$, $\Pr(L, L') = \Pr(L)\Pr(L')$ and $\forall V_t^i \neq V_{t'}^j$, $\mathbb{L}^{v_t^i} \cap \mathbb{L}^{v_{t'}^j} = \varnothing$, *i.e., there is no hidden confounding.*

Furthermore, we suppose that in every unknown DTDSCM, there exists a known maximal lag between causes and effects, denoted as $\gamma_{max}$. Additionally, we assume that the probability distribution $\Pr$ over the observed variables is positive. Finally, we suppose that each DTDSCM induces a full-time acyclic directed mixed graph (FT-ADMG), a specific type of ADMG (Richardson, 2003), where bidirected dashed arrows represent hidden confounding. Under Assumption 1, the FT-ADMG reduces to a full-time directed acyclic graph (FT-DAG).

**Definition 2** (Full-Time Acyclic Directed Mixed Graph). *Consider a DTDSCM $\mathcal{M}$. The* full-time acyclic directed mixed graph (FT-ADMG) $\mathcal{G} = (\mathbb{V}, \mathbb{E})$ *induced by $\mathcal{M}$ is defined in the following way:*

$$\mathbb{E}^1 := \{X_{t-\gamma} \to Y_t \mid \forall Y_t \in \mathbb{V}, \ X_{t-\gamma} \in \mathbb{X} \subseteq \mathbb{V}_{\leq t} \backslash \{Y_t\}$$
$$\text{such that } Y_t := f^{y_t}(\mathbb{X}, \mathbb{L}^{y_t}) \text{ in } \mathcal{M}\},$$
$$\mathbb{E}^2 := \{X_{t-\gamma} \leftarrow\!\dashrightarrow Y_t \mid \forall Y_t, \forall X_{t-\gamma} \in \mathbb{V} \backslash \{X_{t-\gamma}\}$$
$$\text{such that } \Pr(\mathbb{L}^{x_{t-\gamma}} \cap \mathbb{L}^{y_t}) \neq \Pr(\mathbb{L}^{x_{t-\gamma}})\Pr(\mathbb{L}^{y_t})\},$$

*where $\mathbb{E} = \mathbb{E}^1 \cup \mathbb{E}^2$.*

The findings of this study, when viewed in isolation, do not rely on any additional assumptions. However, following the identification of the direct effect, the estimation phase involves using actual data. If the data consist of time series (e.g., various time series describing a specific patient), a stationarity assumption becomes necessary to satisfy the positivity assumption. Conversely, the stationarity assumption is not required when working with spatio-temporal or cohort data.

In the context of DTDSCM and FT-ADMG, both macro and micro causal effect questions can be posed (Ferreira & Assaad, 2024b). A macro causal effect refers to the influence of an entire time

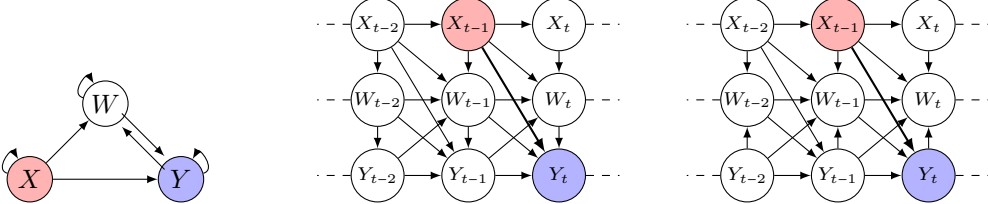

Figure 1: An SCG in (a) with two compatible FT-ADMGs in (b) and (c). Each pair of red and blue vertices represents the micro direct effect we are interested in. Both the controlled direct effect and the natural direct effect are not identifiable using the conditions given in this paper.

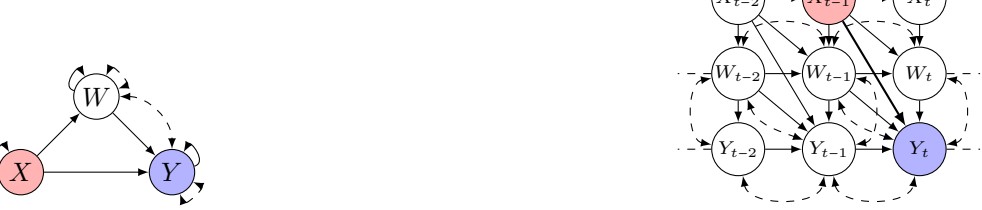

Figure 2: An SCG in (a) with a compatible FT-ADMG in (b). Each pair of red and blue vertices represents the micro direct effect we are interested in. Both the controlled direct effect and the natural direct effect are not identifiable using the conditions given in this paper.

series (or multiple time series) on another series, whereas a micro causal effect focuses on the influence of specific time points (or multiple time points) on other time points. This work specifically focuses on micro causal effects, with a particular emphasis on micro direct effects. In a linear setting (where all functions in $\mathbb{F}$ are linear), the micro direct effect of $X_{t-\gamma}$ on $Y_t$ is represented by the path coefficient (Wright, 1920; 1921) of $X_{t-\gamma}$ in $f^{y_t}$ within the DTDSCM. However, in a non-parametric setting, defining the direct effect becomes more complex. The literature distinguishes two types of non-parametric direct effects, the first being the controlled direct effect, which is defined as follows.

**Definition 3** (Average Controlled Micro Direct Effect (Pearl, 2001)). *Given a FT-ADMG $\mathcal{G} = (\mathbb{V}, \mathbb{E})$, $Y_t, X_{t-\gamma} \in \mathbb{V}$ and $\mathbb{Z} = Parents(Y_t, \mathcal{G}) \backslash \{X_{t-\gamma}\}$. The controlled micro direct effect of changing $X_{t-\gamma}$ from $x$ to $x'$ on $Y_t$ while keeping $\mathbb{Z}$ at value $\mathbb{z}$ is defined as*

$$CDE(X_{t-\gamma}^{x,x'}, Y_t, \mathbb{z}) = E(Y_t \mid do(X_{t-\gamma} = x'), do(\mathbb{Z} = \mathbb{z})) - E(Y_t \mid do(X_{t-\gamma} = x), do(\mathbb{Z} = \mathbb{z})).$$

In general, average controlled micro direct effects cannot be used for effect decomposition and therefore are not useful in mediation analysis. Specifically, the difference between the total effect and the average controlled micro direct effect cannot typically be interpreted as an indirect effect (Robins & Greenland, 1992; Pearl, 2001; Kaufman et al., 2004; Vanderweele, 2011). This issue can be solved when we set all values of parents of $Y_t$ to whatever value they would have obtained under $do(X_{t-\gamma} = x)$. This is known as the natural direct effect (Pearl, 2001) or the pure direct effect (Robins & Greenland, 1992)).

**Definition 4** (Average Natural Micro Direct Effect (Pearl, 2001)). *Given a FT-ADMG $\mathcal{G} = (\mathbb{V}, \mathbb{E})$, $Y_t, X_{t-\gamma} \in \mathbb{V}$ and $\mathbb{Z} = Parents(Y_t, \mathcal{G}) \backslash \{X_{t-\gamma}\}$. The natural micro direct effect of changing $X_{t-\gamma}$ from $x$ to $x'$ on $Y_t$ is defined as*

$$NDE(X_{t-\gamma}^{x,x'}, Y_t) = E(Y_t \mid do(X_{t-\gamma} = x'), do(\mathbb{Z} = \mathbb{z}_x)) - E(Y_t \mid do(X_{t-\gamma} = x))$$

*where $\mathbb{z}_x$ is the counterfactual value of $\mathbb{Z}$ under the setting $X_{t-\gamma} = x$.*

Note that in Pearl (2014), $\mathbb{Z}$ is not the parents of $Y_t$ but the set of intermediate variables (*i.e.*, mediators) between $X_{t-\gamma}$ and $Y_t$. In this work we will focus on the set of parents of the effect, as in Pearl (2001), since it allows a better parallel with the definition of the average controlled micro direct effect that we adopted.

There has been substantial work on identifying both average controlled and average natural direct effects from observational data using ADMGs, and by extension, FT-ADMGs. These direct effects

are considered identifiable if they can be expressed solely in terms of conditional probabilities and expectations over observed variables. Notably, the task of identifying these effects can, in some cases, be fully reduced to identifying interventional distributions alone. For instance, in the case of the average controlled direct effect, identification relies entirely on identifying the interventional distribution $\Pr(y_t \mid \mathrm{do}(X_{t-\gamma} = x), \mathrm{do}(\mathbb{Z} = \mathbb{z}))$. Meanwhile, for the average natural direct effect, it partially depends on identifying the interventional distribution $\Pr(y_t \mid \mathrm{do}(X_{t-\gamma} = x))$ and additionally requires identifying a counterfactual term. However, as demonstrated by Pearl (2001) and as we will show in Section 5, even this counterfactual term can sometimes be identified by relating it back to specific interventional distributions. The do-calculus, introduced by Pearl (1995) provides a symbolic machinery that can be used to identify interventional distributions. It compromises three rules, each establishing specific graphical criteria that dictate when substitutions can be applied to an interventional distribution. The most important aspect of the do-calculus is that it is sound and complete: any interventional distribution is identifiable if and only if there exists a sequence of the rules of the do-calculus that transforms the interventional distribution into an expression containing only conditional probabilities and expectations over observed variables (Pearl, 1995; Shpitser & Pearl, 2006; Huang & Valtorta, 2006). The three rules of the do-calculus are as follows:

**R1:** $\Pr(\mathbb{Y} = y \mid \mathrm{do}(\mathbb{Z} = \mathbb{z}), \mathbb{X} = \mathbb{x}, \mathbb{W} = \mathbb{w}) = \Pr(\mathbb{Y} = y \mid \mathrm{do}(\mathbb{Z} = \mathbb{z}), \mathbb{W} = \mathbb{w})$      if $\mathbb{Y} \perp\!\!\!\perp_{\mathcal{G}_{\overline{\mathbb{Z}}}} \mathbb{X} \mid \mathbb{Z}, \mathbb{W}$

**R2:** $\Pr(\mathbb{Y} = y \mid \mathrm{do}(\mathbb{Z} = \mathbb{z}), \mathrm{do}(\mathbb{X} = \mathbb{x}), \mathbb{W} = \mathbb{w}) = \Pr(\mathbb{Y} = y \mid \mathrm{do}(\mathbb{Z} = \mathbb{z}), \mathbb{X} = \mathbb{x}, \mathbb{W} = \mathbb{w})$    if $\mathbb{Y} \perp\!\!\!\perp_{\mathcal{G}_{\overline{\mathbb{Z}}\underline{\mathbb{X}}}} \mathbb{X} \mid \mathbb{Z}, \mathbb{W}$

**R3:** $\Pr(\mathbb{Y} = y \mid \mathrm{do}(\mathbb{Z} = \mathbb{z}), \mathrm{do}(\mathbb{X} = \mathbb{x}), \mathbb{W} = \mathbb{w}) = \Pr(\mathbb{Y} = y \mid \mathrm{do}(\mathbb{Z} = \mathbb{z}), \mathbb{W} = \mathbb{w})$      if $\mathbb{Y} \perp\!\!\!\perp_{\mathcal{G}_{\overline{\mathbb{Z}\mathbb{X}(\mathbb{W})}}} \mathbb{X} \mid \mathbb{Z}, \mathbb{W}$

where $\mathbb{X}(\mathbb{W})$ is the set of vertices in $\mathbb{X}$ that are non-ancestors of any vertex in $\mathbb{W}$ in the mutilated graph $\mathcal{G}_{\overline{\mathbb{Z}}}$.

The interventional distribution is said to be identifiable by adjustment when it can be expressed as a sum of probabilities of the response conditioned on the treatment and a subset of observed variables $\mathbb{W}$, weighted by the probability of observing those values of $\mathbb{W}$, *i.e.*, $\sum_{\mathbb{w}} \Pr(Y_t = y \mid X_{t-\gamma} = x, \mathbb{W} = \mathbb{w})\Pr(\mathbb{W} = \mathbb{w})$.

However, it is challenging for practitioners in fields like epidemiology to provide or validate a FT-ADMG, as this requires detailed knowledge of temporal dynamics at every time point, which can be both complex and impractical. Moreover, causal discovery algorithms (Spirtes et al., 2000) (*i.e.*, algorithms which infer a causal graph from observational data under strong untestable assumptions) are still largely ineffective on real-world data (Aït-Bachir et al., 2023). Therefore, it is often more feasible to use summary causal graphs, which simplify the causal structure into a more manageable form. This allows practitioners to focus on the key pathways and relationships. For instance, when studying the long-term effects of smoking on lung disease, epidemiologists can outline the main causal pathways—like smoking leading to lung damage—without needing to detail the exact timing of each intermediate step.

**Definition 5** (Summary Causal Graph with possible latent confounding). *Consider an FT-ADMG* $\mathcal{G} = (\mathbb{V}, \mathbb{E})$. *The* summary causal graph (SCG) $\mathcal{G}^s = (\mathbb{S}, \mathbb{E}^s)$ *compatible with* $\mathcal{G}$ *is defined in the following way:*

$$\mathbb{S} := \{V^i = (V^i_{t_0}, \cdots, V^i_{t_{max}}) \mid \forall i \in [1, d]\},$$
$$\mathbb{E}^{s1} := \{X \to Y \mid \forall X, Y \in \mathbb{S}, \ \exists t' \le t \in [t_0, t_{max}] \quad \textit{such that } X_{t'} \to Y_t \in \mathbb{E}\},$$
$$\mathbb{E}^{s2} := \{X \leftarrow\!\!\dashrightarrow Y \mid \forall X, Y \in \mathbb{S}, \ \exists t', t \in [t_0, t_{max}] \quad \textit{such that } X_{t'} \leftarrow\!\!\dashrightarrow Y_t \in \mathbb{E}\}.$$

*where* $\mathbb{E}^s = \mathbb{E}^{s1} \cup \mathbb{E}^{s2}$.

Figures 1-4 present many examples of SCGs with compatible FT-ADMGs. The abstraction of SCGs implies that, while a given FT-ADMG corresponds to exactly one SCG, multiple FT-ADMGs can be compatible with the same SCG. As a result, the do-calculus as introduced by Pearl (1995), is not directly applicable to SCGs to identify micro causal effects (Ferreira & Assaad, 2024b). The objective of this paper is to establish specific graphical conditions within an SCG that guarantee the existence of a sequence of do-calculus rules that can be applied to any FT-ADMG compatible with the SCG, yielding an identical expression involving only conditional probabilities and expectations over observed variables. A key insight about SCGs that aids in identifying these conditions is that, although there may not be a strict one-to-one correspondence between the set of parents of a vertex $Y$ in an SCG and the set of parents of $Y_t$ in any particular FT-ADMG, a precise mapping is possible

when considering the set of all potential parents across all FT-ADMGs compatible with the SCG. Specifically, this set includes vertices that act as parents in at least one FT-ADMG that aligns with the SCG. Formally, this set of possible parents is defined as follows.

**Definition 6** (Possible Parents). *Given an SCG $\mathcal{G}^s = (\mathbb{S}, \mathbb{E}^s)$, a maximal lag $\gamma_{max}$ and a vertex $Y \in \mathbb{S}$. The set of possible parents of the temporal vertex $Y_t$ is the set of temporal vertices (i.e.vertices in compatible FT-ADMGs) which are parents of $Y_t$ in at least one compatible FT-ADMG. It is written $PP(Y_t)$ and is defined as*

$$PP(Y_t) = \{P_{t-\gamma} \mid P \in Parents(Y, \mathcal{G}^s) \backslash \{Y\}, \gamma \in [0, \gamma_{max}]\} \cup \{Y_{t-\gamma} \mid \gamma \in (0, \gamma_{max}]\} * \mathbb{1}_{Y \in Parents(Y, \mathcal{G}^s)}.$$

We close this section with the following property of possible parents, which shows that it is possible to replace the set of parents in the definitions of the average controlled micro direct effect and the average natural micro direct effect by the set of potential parents.

**Property 1.** *Given a FT-ADMG $\mathcal{G} = (\mathbb{V}, \mathbb{E})$, $Y_t \in \mathbb{V}$ and $\mathbb{Z} \supseteq Parents(Y_t, \mathcal{G})$ such that $Y_t \notin \mathbb{Z}$, the following equality holds for all $\mathbb{z}$:*

$$\Pr(Y_t = y \mid do(Parents(Y_t, \mathcal{G}) = \mathbb{z}\mid_{Parents(Y_t, \mathcal{G})})) = \Pr(Y_t = y \mid do(\mathbb{Z} = \mathbb{z})).$$

*This equality holds specifically for $\mathbb{Z} = PP(Y_t)$ and note that there exists a compatible FT-ADMG $\mathcal{G}'$ in which $PP(Y_t) = Parents(Y_t, \mathcal{G}')$.*

This property results directly from R3 of the do-calculus. All formal proof can be found in the supplementary materials.

## 4 IDENTIFYING CONTROLLED MICRO DIRECT EFFECTS IN SCGs

As stated in the previous section, the average controlled micro direct effect of interest is the one where we intervene on all parents of $Y_t$. In order to identify this effect, in particular using R2 of the do-calculus, one can find a set of variables of $Y_t$ which d-separates $Y_t$ from its parents in $\mathcal{G}_{\overline{Parents(Y_t, G)}}$ for all compatible FT-ADMG $\mathcal{G}$. There are two significant challenges in this process. The first challenge arises when there exists a cycle involving $Y$ and another vertex in the SCG. For example, consider the SCG in Figure 1(a), where Assumption 1 holds. In this case, the cycle between $Y$ and $W$ implies the presence of a compatible FT-ADMG $\mathcal{G}$, as depicted in Figure 1(b), in which $W_t \in Parents(Y_t, \mathcal{G})$. In this FT-ADMG we would like to ultimately remove all the $do(\cdot)$ operators in the interventional distribution $\Pr(Y_t = y \mid do(W_t = w), do(Y_{t-1} = y'), do(X_{t-1} = x), do(W_{t-1} = w'))$ but for the sake of illustration let us only focus on removing the $do(\cdot)$ on $W_t$. Notice that in this FT-ADMG, we have $Y_t \perp\!\!\!\perp_{\mathcal{G}_{\overline{Y_{t-1}X_{t-1}W_{t-1}}W_t}} W_t \mid Y_{t-1}, X_{t-1}, W_{t-1}$, which means that by using using R2 of the do-calculus, it is possible to write the considered interventional distribution as $\Pr(Y_t = y \mid W_t = w, do(Y_{t-1} = y'), do(X_{t-1} = x), do(W_{t-1} = w'))$. However, there exists another compatible FT-ADMG $\mathcal{G}'$, shown in Figure 1(c) where $W_t \leftarrow Y_t$ and thus it is impossible to use R2 to remove the $do(\cdot)$ operator on $W_t$ in the same interventional distribution since $\forall \mathbb{W}$, $Y_t \not\perp\!\!\!\perp_{\mathcal{G}'_{\overline{Y_{t-1}X_{t-1}W_{t-1}}W_t}} W_t \mid \mathbb{W}$. This indicates that the relationship between $W_t$ and $Y_t$ is ambiguous when only the SCG is considered, preventing the removal of the $do(\cdot)$ operator on $W_t$ in the considered interventional distribution. The second challenge occurs when hidden confounding exists between $Y$ and another vertex. For instance, consider the SCG in Figure 2(a), in this scenario, by examining the compatible FT-ADMG $\mathcal{G}$ in Figure 2(b), it becomes clear that as in the previous example, it is necessary to remove the $do(\cdot)$ on $W_t$ in the interventional distribution $\Pr(Y_t = y \mid do(W_t = w), do(Y_{t-1} = y'), do(X_{t-1} = x), do(W_{t-1} = w'))$ as $W_t \in Parents(Y_t, \mathcal{G})$. However, hidden confounding exists between $W_t$ and $Y_t$, making R2 and R3 of the do-calculus inapplicable since $\forall \mathbb{W}$, $Y_t \not\perp\!\!\!\perp_{\mathcal{G}_{\overline{Y_{t-1}X_{t-1}W_{t-1}}W_t}} W_t \mid \mathbb{W}$ and $Y_t \not\perp\!\!\!\perp_{\mathcal{G}_{\overline{Y_{t-1}X_{t-1}W_{t-1}}W_t(\mathbb{W})}} W_t \mid \mathbb{W}$[1]. Nevertheless, as shown in the following theorem, it is possible to identify the controlled micro direct effect in any SCG where no cycle involving $Y$ and other vertices exists and where hidden confounding between $Y$ and its ancestors is absent.

---

[1] The subgraph formed by $Y_t$ and $W_t$ in Figure 2(b) is known as an bow arc graph and when such a structure exists, the interventional distribution $\Pr(Y_t = y \mid do(W_t = w))$ is known to be non-identifiable (Shpitser & Pearl, 2006)

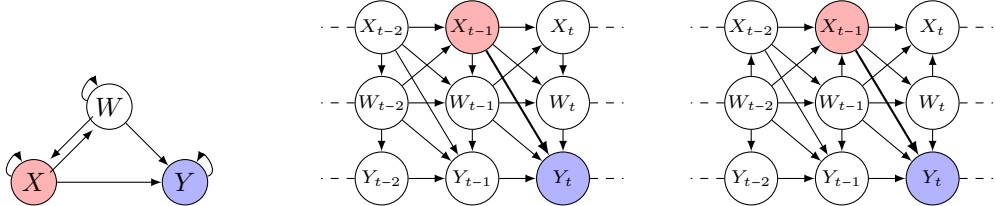

Figure 3: An SCG in (a) with two compatible FT-ADMGs in (b) and (c). Each pair of red and blue vertices represents the micro direct effect we are interested in. The controlled direct effect is identifiable according to our condition but the natural direct effect is not.

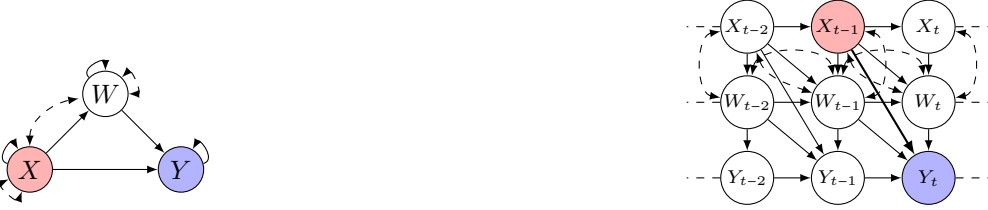

Figure 4: An SCG in (a) with a compatible FT-ADMG in (b). Each pair of red and blue vertices represents the micro direct effect we are interested in. The controlled direct effect is identifiable according to our condition but the natural direct effect is not.

**Theorem 1.** *Given an SCG $\mathcal{G}^s = (\mathbb{S}, \mathbb{E}^s)$, $Y \in \mathbb{S}$ and $X_{t-\gamma} \in PP(Y_t)$. The controlled direct effect of changing $X_{t-\gamma}$ from $x$ to $x'$ on $Y_t$ is identifiable if $Scc(Y, \mathcal{G}^s) \subseteq \{Y\}$ and there does not exist a bidirected dashed arrow between $Y$ and one of its ancestors (i.e. $\nexists Z \in Ancestors(Y, \mathcal{G}^s)$, $Z \leftarrow\!\dashrightarrow Y$) and we have:*

$$CDE(X_{t-\gamma}^{x,x'}, Y_t, \mathbb{z}) = E(Y_t \mid X_{t-\gamma} = x', \mathbb{Z} = \mathbb{z}) - E(Y_t \mid X_{t-\gamma} = x, \mathbb{Z} = \mathbb{z})$$

*where $\mathbb{Z} = PP(Y_t) \backslash \{X_{t-\gamma}\}$.*

For illustration, we give in Figure 5 three SCGs where the average controlled micro directed effect is identifiable by Theorem 1.

Theorem 1 offers sufficient conditions for identifying the average controlled micro direct effect supposing a general unknown FT-ADMG. It's important to note, however, that these conditions are not necessary in general. Interestingly, under Assumption 1 (*i.e.*, supposing an unknown FT-DAG) and by only considering identifiability by adjustment, these conditions do become necessary, as demonstrated in the following proposition.

**Proposition 1.** *Given an SCG $\mathcal{G}^s = (\mathbb{S}, \mathbb{E}^s)$, $Y \in \mathbb{S}$ and $X_{t-\gamma} \in PP(Y_t)$. Under Assumption 1, if $Scc(Y, \mathcal{G}^s) \nsubseteq \{Y\}$ then the controlled direct effect of changing $X_{t-\gamma}$ from $x$ to $x'$ on $Y_t$ is not identifiable by adjustment.*

## 5 IDENTIFYING NATURAL MICRO DIRECT EFFECTS IN SCGS

Pearl (2014) provided sufficient conditions to reframe the problem of identifying the average natural direct effect from one involving counterfactuals to one based on interventions, specifically using the $do()$ operator. The formal result addressed cases where $\mathbb{Z}$ represents all mediators. It was informally observed that the same conditions apply when $\mathbb{Z}$ includes all parent variables. Since this paper focuses on parent variables, we restate and formalize this result in the context of parents.

**Lemma 1.** *Given an SCG $\mathcal{G}^s = (\mathbb{S}, \mathbb{E}^s)$, let $Y \in \mathbb{S}$, $X_{t-\gamma} \in PP(Y_t)$. If $Scc(Y, \mathcal{G}^s) \subseteq \{Y\}$ and there is no bidirected dashed arrow between $Y$ and its ancestors then the natural direct effect of changing $X_{t-\gamma}$ from $x$ to $x'$ on $Y_t$, $NDE(X_{t-\gamma}^{x,x'}, Y_t)$, is identifiable from the SCG if*

- $\Pr(Y_t = y \mid do(X_{t-\gamma} = x), do(\mathbb{Z} = \mathbb{z}))$ *is identifiable from the SCG, and*

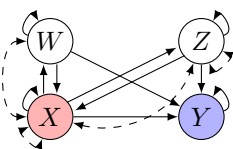 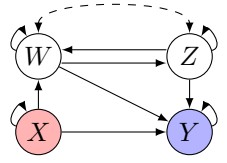 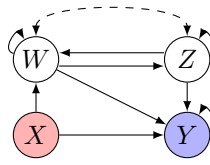

Figure 5: Three SCGs. Each pair of red and blue vertices represents the micro direct effect we are interested in. The controlled direct effect is identifiable in all these SCGs using the conditions given in this paper. The natural direct effect is not identifiable in (a), identifiable if $\gamma = \gamma_{max}$ in (b) and identifiable for all $\gamma$ in (c).

- $\Pr(\mathbb{Z} = \mathbb{z} \mid do(X_{t-\gamma} = x))$ *is identifiable from the SCG.*

*where* $\mathbb{Z} = PP(Y_t)\backslash\{X_{t-\gamma}\}$.

Notice that Lemma 1 shows that the the identifiability of the average natural micro direct effect can be reduced to the identifiability of the average controlled micro direct effect and the identifiability of the interventional distribution $\Pr(\mathbb{Z} = \mathbb{z} \mid \text{do}(X_{t-\gamma} = x))$. The first identifiability can be directly given by the result of the previous section which means that we need to impose the constraints given for the average controlled micro direct effect to the average natural micro direct effect. In addition, we need to add some conditions to ensure the second identifiability. The SCG in Figure 3(a) represents a case where this second identifiability cannot be achieved. To see this, consider $\Pr(W_{t-1} = w, W_t = w', Y_{t-1} = y \mid \text{do}(X_{t-1} = x))$ and notice that, similarly to Figure 1, because of the cycle $X \rightleftarrows W$ in the SCG, the $\text{do}(\cdot)$ operator cannot be removed using the same sequence of do-calculus rules in the first compatible FT-ADMG in Figure 3(b) in which $X_{t-1} \rightarrow W_{t-1}$ and in the second compatible FT-ADMG in Figure 3(c) in which $X_{t-1} \leftarrow W_{t-1}$. The SCG in Figure 4(a) represents another case where this second identifiability cannot be achieved due to hidden confounding. To see this, consider $\Pr(W_{t-1} = w, W_t = w', Y_{t-1} = y \mid \text{do}(X_{t-1} = x))$ and notice that, similarly to Figure 2, because of hidden confounding $X \leftarrow\!\!\dashrightarrow W$ in the SCG, the $\text{do}(\cdot)$ operator cannot be removed in this interventional distribution in the compatible FT-ADMG in Figure 4(b) in which $X_{t-1} \leftarrow\!\!\dashrightarrow W_{t-1}$.

**Theorem 2.** *Given an SCG $\mathcal{G}^s$, the natural direct effect of changing $X_{t-\gamma}$ from $x$ to $x'$ on $Y_t$, $NDE(X_{t-\gamma}^{x,x'}, Y_t)$, is identifiable from the SCG if*

1. $Scc(Y, \mathcal{G}^s) \subseteq \{Y\}$,

2. $Scc(X, \mathcal{G}^s) \subseteq \{X\}$,

3. $PP(X_{t-\gamma}) \cap PP(Y_t) = \varnothing$,

4. $\nexists Z \in Ancestors(Y, \mathcal{G}^s)$, $Z \leftarrow\!\!\dashrightarrow Y$, *and*

5. $\nexists Z \in Ancestors(Y, \mathcal{G}^s)$, $Z \leftarrow\!\!\dashrightarrow X$.

*When these conditions are satisfied, we have:*

$$NDE(X_{t-\gamma}^{x,x'}, Y_t) = \sum_{\mathbb{z}} \left( CDE(X_{t-\gamma}^{x,x'}, Y_t, \mathbb{z}) \sum_{\mathbb{a}} \Pr(\mathbb{Z} = \mathbb{z} \mid X_{t-\gamma} = x, \mathbb{A} = \mathbb{a})\Pr(\mathbb{A} = \mathbb{a}) \right)$$

*where* $\mathbb{Z} = PP(Y_t)\backslash\{X_{t-\gamma}\}$, $\mathbb{A} = PP(X_{t-\gamma})$ *and a formula for* $CDE(X_{t-\gamma}^{x,x'}, Y_t, \mathbb{z})$ *is given in Theorem 1.*

Note that Condition 3 implies either having no self-loop on $X$ in the SCG or choosing $\gamma = \gamma_{max}$. As an illustration, Figure 5(b) presents an SCG where $NDE(X_{t-\gamma_{max}}^{x,x'}, Y_t)$ is identifiable, though this identifiability does not hold when $\gamma_{max}$ is replaced by $\gamma < \gamma_{max}$. In contrast, Figure 5(c) shows an SCG where $NDE(X_{t-\gamma}^{x,x'}, Y_t)$ is identifiable for all $\gamma$.

## 6 REAL WORLD EXAMPLE

In the following, we give one example in epidemiology (Figure 6) which illustrates a scenario where our theorem can be applicable.

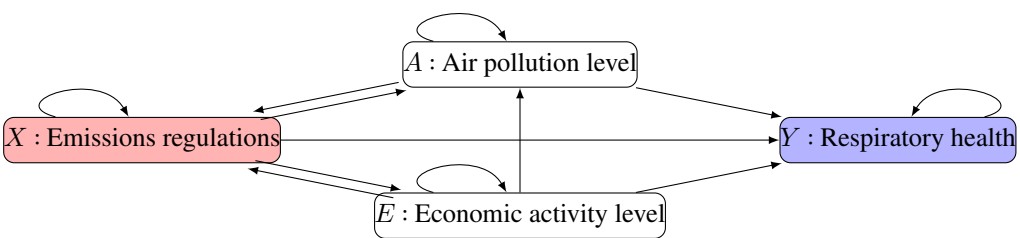

Figure 6: Example in epidemiology

Suppose we want to assess the effect of introducing stricter emissions regulations ($X$) in a city at time $t-1$ on the respiratory health of residents ($Y$) at time $t$. There are at least two confounders between the treatment and outcome:

- Economic activity level ($E$): it affects both pollution levels (which, in turn, impact respiratory health) and the likelihood of implementing emissions regulations. The economic activity level can have a lagged causal relationship with the treatment, or potentially an instantaneous effect, depending on the sampling frequency.
- Air pollution level ($A$): it affects respiratory health outcomes and the likelihood of emissions regulations. Similarly to the first confounder, the relation between air pollution and $X$ or $Y$ can be instantaneous, depending on the sampling frequency.

At the same time, these two confounders can act as mediators between $X$ and $Y$ as regulations can effect the economy and the air pollution level. At the SCG level, if we disregard the time dimension, we obtain a cyclic relationship between each of the two confounders and the treatment. Therefore, these confounders complicate the estimation of the causal effect of emissions regulations on respiratory health.

Assuming that there are no other confounders, the micro CDE is identifiable using Theorem 1, however the micro NDE is not identifiable using Theorem 2.

## 7 CONCLUSION

In this paper, we addressed both average controlled and average natural micro direct effects, providing theoretical results that extend existing work in causal inference using summary causal graphs. By accounting for cycles, hidden confounders and allowing for a non-parametric setting, our results contribute to a broader applicability of causal inference techniques in real-world settings where full causal specification is impractical. Our findings are particularly relevant in fields such as epidemiology, where accurate measurement of direct effects is crucial for informing public health interventions and policy decisions.

In future work, it will be important to validate whether our conditions hold when focusing on other forms of average controlled and average natural direct effects, such as those that emphasize mediators rather than parents. Additionally, it would be valuable to establish necessary and sufficient conditions for average controlled direct effects that extend beyond simple adjustment. Another future direction would be to derive necessary and sufficient conditions for identifying average controlled direct effects in the presence of hidden confounding, as well as for average natural direct effects. However, we believe that this is more challenging to achieve compared to average controlled direct effects without hidden confounding. Now that total and direct effects are well understood in the context of summary causal graphs, it would also be interesting to explore the identification of indirect effects and, more broadly, path-specific effects within summary causal graphs. Finally, we plan to apply the findings of this work to real-world applications, particularly in the field of epidemiology.

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

APPENDIX

## A  PROOFS

*Property 1.* Take $Z_{t_Z} \in \mathbb{Z} \backslash Parents(Y_t, \mathcal{G})$ and notice that $Y_t \perp\!\!\!\perp_{\mathcal{G}_{\overline{\mathbb{Z}}}} Z_{t_Z} \mid Parents(Y_t, \mathcal{G})$. Indeed, $Parents(Y_t, \mathcal{G}) \subseteq \mathbb{Z}$ so in $\mathcal{G}_{\overline{\mathbb{Z}}}$, $Parents(Y_t, \mathcal{G})$ have no parents and thus cannot be colliders nor descendants of colliders. Therefore, in $\mathcal{G}_{\overline{\mathbb{Z}}}$, an active path starting with $Y_t$ must be of the form $\langle Y_t \to \cdots \to \rangle$ or $\langle Y_t \leftarrow\!\!\!-\!\!\!\to \cdot \to \cdots \to \rangle$ and an active path ending with $Z_{t_Z}$ must be of the form $\langle \leftarrow \cdots \leftarrow Z_{t_Z} \rangle$. Clearly, there is no active path from $Y_t$ to $Z_{t_Z}$ in $\mathcal{G}_{\overline{\mathbb{Z}}}$ and the d-separation holds. Thus, using Rule 3 of the do-calculus (Pearl, 1995) we get $\Pr(Y_t = y \mid \mathrm{do}(Parents(Y_t, \mathcal{G}) = \mathbb{p})) = \Pr(Y_t = y \mid \mathrm{do}(\mathbb{Z} = \mathbb{z}))$. □

*Theorem 1.* Using Property 1 one can replace $\mathbb{Z} = Parents(Y_t, \mathcal{G}) \backslash \{X_{t-\gamma}\}$ by $\mathbb{Z} = PP(Y_t) \backslash \{X_{t-\gamma}\}$ in Definition 3. Suppose that $Scc(Y, \mathcal{G}^s) \subseteq \{Y\}$ and that there does not exist a bidirected dashed arrow between $Y$ and one of its ancestors in $\mathcal{G}^s$. Let us show that in every compatible FT-ADMG $\mathcal{G}$, we have $Y_t \perp\!\!\!\perp_{\mathcal{G}_{\underline{PP(Y_t)}}} PP(Y_t)$ in order to use R2 of the do-calculus to write $\Pr(Y_t = y \mid \mathrm{do}(PP(Y_t) = \mathbb{p}))$ as $\Pr(Y_t = y \mid PP(Y_t) = \mathbb{p})$.

Let $\mathcal{G}$ be a compatible FT-ADMG and $Z_{t_Z} \in PP(Y_t)$ and let us consider a path $\pi$ from $Z_{t_Z}$ to $Y_t$ in $\mathcal{G}_{\underline{PP(Y_t)}}$.

- $\pi$ cannot end with a right arrow (*i.e.*, $\pi \neq \langle Z_{t_Z} \cdots \to Y_t \rangle$) as no arrow going in $Y_t$ exists in $\mathcal{G}_{\underline{PP(Y_t)}}$.

- $\pi$ can end with a left arrow (*i.e.*, $\pi = \langle Z_{t_Z} \cdots \leftarrow Y_t \rangle$) but it cannot contain only left arrows (*i.e.*, $\pi \neq \langle Z_{t_Z} \leftarrow \cdots \leftarrow Y_t \rangle$) as if $X = Y$, this would imply having causal arrows going backwards in time which is forbidden or if $X \neq Y$, this would imply $X \in Descendants(Y, \mathcal{G}^s)$ and thus $X \in Scc(Y, \mathcal{G}^s)$ which contradicts the assumption. Therefore, there exists a collider in $\pi$ ($\pi = \langle Z_{t_Z} \cdots \to W_{t_W} \leftarrow \cdots \leftarrow Y_t \rangle$ or $\pi = \langle Z_{t_Z} \cdots \leftarrow\!\!\!-\!\!\!\to W_{t_W} \leftarrow \cdots \leftarrow Y_t \rangle$) and $\pi$ is blocked.

- If $\pi$ ends with a bidirected dashed arrow ($\pi = \langle Z_{t_Z} \cdots U_{t_U} \leftarrow\!\!\!-\!\!\!\to Y_t \rangle$) then by assumption,$U_{t_U}$ is not a possible ancestor of $Y_t$ so since $Z_{t_Z} \in PP(Y_t)$, the remaining arrows cannot all be left arrows ($\pi \neq \langle Z_{t_Z} \leftarrow \cdots \leftarrow U_{t_U} \leftarrow\!\!\!-\!\!\!\to Y_t \rangle$). Therefore, there exists a collider in $\pi$ (*i.e.*, $\pi = \langle Z_{t_Z} \cdots \to W_{t_W} \leftarrow \cdots \leftarrow U_{t_U} \leftarrow\!\!\!-\!\!\!\to Y_t \rangle$ or $\pi = \langle Z_{t_Z} \cdots \leftarrow\!\!\!-\!\!\!\to W_{t_W} \leftarrow \cdots \leftarrow U_{t_U} \leftarrow\!\!\!-\!\!\!\to Y_t \rangle$) and $\pi$ is blocked.

Therefore, all path between $Y_t$ and $PP(Y_t)$ are blocked in $\mathcal{G}_{\underline{PP(Y_t)}}$, *i.e.*, $Y_t \perp\!\!\!\perp_{\mathcal{G}_{\underline{PP(Y_t)}}} PP(Y_t)$. □

*Proposition 1.* Firstly using Property 1 one can replace $\mathbb{Z} = Parents(Y_t, \mathcal{G})$ by $\mathbb{Z} = PP(Y_t)$ in Definition 3. Let us suppose that $Scc(Y, \mathcal{G}^s) \not\subseteq \{Y\}$, then there exists a cycle on $Y$ other than the self-loop in the SCG, *i.e.*$\exists C \in Cycle(Y, \mathcal{G}^s) \backslash \{\langle Y \rangle\}$. Let us write $C = \langle V^1, \cdots, V^n \rangle$ with $n \geq 3$, $V^1 = V^n = Y$, $\forall 1 \leq i < n, V^i \neq V^{i+1}$ and $V^i \to V^{i+1}$ or $V^i \rightleftarrows V^{i+1}$. Then, $\langle V^{n-1}, \cdots, V^1 \rangle$ is a active non-direct path containing only descendants of $Y$ so according to Theorem 1 of Ferreira & Assaad (2024a) one cannot be certain to block every non-direct from $V_t^{n-1}$ (which is a potential parent of $Y_t$) to $Y_t$ without risking to induce a bias by adjusting on descendants of $Y_t$ and thus the controlled direct effect is not identifiable by adjustment. □

*Proof.* Lemma 1 As in Pearl (2014), the first term $E(Y_t \mid \mathrm{do}(X_{t-\gamma} = x'), \mathrm{do}(\mathbb{Z} = \mathbb{z}_x))$ in $NDE(X_{t-\gamma}^{x,x'}, Y_t)$ can be written as $\sum_{\mathbb{z}} E(Y_t \mid \mathrm{do}(X_{t-\gamma} = x'), \mathrm{do}(\mathbb{Z} = \mathbb{z}), \mathbb{Z}_x = \mathbb{z})\Pr(\mathbb{Z} = \mathbb{z} \mid \mathrm{do}(X_{t-\gamma} = x))$. Using $Y_t \perp\!\!\!\perp_{\mathcal{G}_{\underline{PP(Y_t)}}} PP(Y_t)$ which was shown in the proof of Theorem 1 one can obtain $E(Y_t \mid \mathrm{do}(X_{t-\gamma} = x'), \mathrm{do}(\mathbb{Z} = \mathbb{z}_x)) = \sum_{\mathbb{z}} E(Y_t \mid \mathrm{do}(X_{t-\gamma} = x'), \mathrm{do}(\mathbb{Z} = \mathbb{z}))\Pr(\mathbb{Z} = \mathbb{z} \mid \mathrm{do}(X_{t-\gamma} = x))$. The second term $E(Y_t \mid \mathrm{do}(X_{t-\gamma} = x))$ in $NDE(X_{t-\gamma}^{x,x'}, Y_t)$ can be modified similarly since using the law of composition (Pearl, 2009, Chapter 7, Property 1) $E(Y_t \mid \mathrm{do}(X_{t-\gamma} = x)) = E(Y_t \mid \mathrm{do}(X_{t-\gamma} = x), \mathrm{do}(\mathbb{Z} = \mathbb{z}_x))$. Therefore, if $\Pr(Y_t = y \mid \mathrm{do}(X_{t-\gamma} = x), \mathrm{do}(\mathbb{Z} = \mathbb{z}))$ and $\Pr(\mathbb{Z} = \mathbb{z} \mid \mathrm{do}(X_{t-\gamma} = x))$ are identifiable from the SCG then $NDE(X_{t-\gamma}^{x,x'}, Y_t)$ is identifiable. □

*Theorem2.* Conditions 1 and 4 allow to use lemma 1. Therefore, it is sufficient to identify $\Pr(Y_t = y \mid \mathrm{do}(X_{t-\gamma} = x), \mathrm{do}(\mathbb{Z} = \mathbb{z}))$ and $\Pr(\mathbb{Z} = \mathbb{z} \mid \mathrm{do}(X_{t-\gamma} = x))$. Theorem 1 states that $\Pr(Y_t = y \mid \mathrm{do}(X_{t-\gamma} = x), \mathrm{do}(\mathbb{Z} = \mathbb{z}))$ is identifiable given Conditions 1 and 4, thus what is left to show is how $\Pr(\mathbb{Z} = \mathbb{z} \mid \mathrm{do}(X_{t-\gamma} = x))$ is identifiable. By the law of total probability, $\Pr(\mathbb{Z} = \mathbb{z} \mid \mathrm{do}(X_{t-\gamma} = x)) = \sum_{\mathbb{a}} \Pr(\mathbb{Z} = \mathbb{z} \mid \mathrm{do}(X_{t-\gamma} = x), = \mathbb{a}) \Pr(\mathbb{A} = \mathbb{a} \mid \mathrm{do}(X_{t-\gamma} = x))$. Notice that because of Conditions 2 and 5 we have for all compatible FT-ADMG $\mathcal{G}$, $\mathbb{A} \perp\!\!\!\perp_{\mathcal{G}_{\overline{X_{t-\gamma}}}} X_{t-\gamma}$ and thus using R3 of the do-calculus $\Pr(\mathbb{A} = \mathbb{a} \mid \mathrm{do}(X_{t-\gamma} = x)) = \Pr(\mathbb{A} = \mathbb{a})$. Let us show that in every compatible FT-ADMG $\mathcal{G}$, $X_{t-\gamma} \perp\!\!\!\perp_{\mathcal{G}_{\underline{X_{t-\gamma}}}} \mathbb{Z} \mid \mathbb{A}$ which will allow to use R2 of the do-calculus and get $\Pr(\mathbb{Z} = \mathbb{z} \mid \mathrm{do}(X_{t-\gamma} = x), \mathbb{A} = \mathbb{a}) = \Pr(\mathbb{Z} = \mathbb{z} \mid X_{t-\gamma} = x, \mathbb{A} = \mathbb{a})$. Let $\mathcal{G}$ be a compatible FT-ADMG and $Z_{t_Z} \in \mathbb{Z}$ and suppose there exists a path $\pi = \langle X_{t-\gamma} \cdots Z_{t_Z} \rangle$ in $\mathcal{G}_{\underline{X_{t-\gamma}}}$.

- $\pi$ cannot start with a right arrow (*i.e.*, $\pi \neq \langle X_{t-\gamma} \rightarrow \cdots Z_{t_Z} \rangle$) as no arrow going out of $X_{t-\gamma}$ exists in $\mathcal{G}_{\underline{X_{t-\gamma}}}$.

- $\pi$ cannot start with a left arrow and be of length 2 (*i.e.*, $\pi \neq \langle X_{t-\gamma} \leftarrow Z_{t_Z} \rangle$) as this contradicts Condition 3.

- $\pi$ can start with a left arrow if it has length at least 2 (*i.e.*, $\pi = \langle X_{t-\gamma} \leftarrow U_{t_U} \cdots Z_{t_Z} \rangle$) as $U_{t_U} \in \mathbb{A}$ and thus $\mathbb{A}$ blocks $\pi$.

- If $\pi$ starts with a bidirected dashed arrow (*i.e.*, $\pi = \langle X_{t-\gamma} \leftarrow\!-\!-\!\rightarrow \cdots Z_{t_Z} \rangle$) then length of $\pi$ cannot be 2 (*i.e.*, $\pi \neq \langle X_{t-\gamma} \leftarrow\!-\!-\!\rightarrow Z_{t_Z} \rangle$) and the remaining arrow cannot all be right arrows ($\pi \neq \langle X_{t-\gamma} \leftarrow\!-\!-\!\rightarrow U_{t_U} \rightarrow \cdots \rightarrow Z_{t_Z} \rangle$) as this would imply $U \in Ancestors(Z, \mathcal{G}^s) \subseteq Ancestors(Y, \mathcal{G}^s)$ which contradicts Condition 5. Therefore, there exists a collider in $\pi$ (*i.e.*, $\pi = \langle X_{t-\gamma} \leftarrow\!-\!-\!\rightarrow U_{t_U} \rightarrow \cdots \rightarrow W_{t_W} \leftarrow \cdots Z_{t_Z} \rangle$ or $\langle X_{t-\gamma} \leftarrow\!-\!-\!\rightarrow U_{t_U} \rightarrow \cdots \rightarrow W_{t_W} \leftarrow\!-\!-\!\rightarrow \cdots Z_{t_Z} \rangle$) which verifies $Descendants(U, \mathcal{G}^s) \supseteq Descendants(W, \mathcal{G}^s)$ and Condition 5 implies $Parents(X, \mathcal{G}^s) \cap Descendants(U, \mathcal{G}^s) = \varnothing$ so $Parents(X, \mathcal{G}^s) \cap Descendants(W, \mathcal{G}^s) = \varnothing$. Thus $\mathbb{A} \cap Descendants(W_{t_W}, \mathcal{G}) = \varnothing$ and $\mathbb{A}$ blocks $\pi$.

All those cases are exhaustive and thus under Conditions 2, 3 and 5, $\mathbb{A}$ is a valid adjustment set to identify $\Pr(\mathbb{Z} = \mathbb{z} \mid \mathrm{do}(X_{t-\gamma} = x))$. Therefore, the 5 conditions together imply the identifiability of $NDE(X_{t-\gamma}^{x,x'}, Y_t)$. $\qquad\square$

