# OpenReview forum: "AVERAGE CONTROLLED AND AVERAGE NATURAL MI-CRO DIRECT EFFECTS IN SUMMARY CAUSAL GRAPHS"
_ICLR.cc/2026/Conference — Submitted to ICLR 2026_

### Official Review · Reviewer_sGFF · 2025-10-28

**Soundness:** 4
**Presentation:** 2
**Contribution:** 4
**Rating:** 6
**Confidence:** 4

**Summary:**

This paper extends the theory of causal effect decomposition to dynamic and potentially cyclic systems. Building on classical notions of controlled and natural direct effects, the authors introduce average controlled micro direct effects and average natural micro direct effects that capture short-lag causal influences within summary causal graphs. The central theoretical contribution is a set of graphical identification conditions establishing when these micro-level effects can be identified nonparametrically, even in the presence of feedback and latent confounders. The framework thereby generalizes existing identification results for nested counterfactuals to a broader class of temporal or cyclic models. The work is mathematically rigorous and conceptually ambitious, with particular relevance for fields such as epidemiology, where quantifying direct causal effects under complex dependencies is essential.

**Strengths:**

1. Ambitious theoretical extension. The paper generalizes classical definitions of controlled and natural direct effects to dynamic and cyclic systems through the “micro” formulation, representing a substantial conceptual step beyond standard acyclic mediation models.
2. Clear linkage to existing theory. The proposed definitions of average controlled and average natural micro direct effects are faithful extensions of established causal inference concepts (Robins & Greenland, Pearl, Avin et al.) into the time-indexed domain.
3. Potential impact on causal mediation in feedback systems. By providing graphical identification conditions for micro direct effects in summary causal graphs, the paper opens new directions for mediation analysis in recurrent and continuous-time causal models.
4. Rigor and completeness. The theoretical development is mathematically precise, and the proofs are carefully structured, offering a thorough and internally consistent foundation for subsequent applied or methodological work.
5. Generality and scope. The results are formulated abstractly enough to encompass a broad class of dynamic systems, including those with feedback and latent confounding, making the framework widely applicable.

**Weaknesses:**

1. Accessibility and clarity.
The paper is technically dense, and the intuition behind the key definitions—especially the natural micro direct effect—is difficult to follow without deep familiarity with classical causal mediation theory. More verbal explanation or illustrative examples would help readers grasp the motivation and implications of the results.
2. Empirical demonstration.
Section 6 offers a well-chosen conceptual example in epidemiology that clarifies the kinds of settings where the theorems could apply, but it stops short of a genuine data analysis or simulation. The example illustrates applicability but does not substantively validate the framework empirically.
3.	Connection to prior identifiability results.
While the paper references Pearl and Avin et al., it could more clearly articulate how its graphical conditions differ from or extend earlier nonparametric identification results for nested counterfactuals. A concise comparative discussion would help situate the contribution relative to existing theory.
4.	Scope and assumptions.
The identifiability results depend on relatively strong structural assumptions about the summary causal graph, and it remains unclear how often those conditions hold in realistic dynamic data. Some discussion of practical plausibility or examples of compliant systems would be useful.
5.	Positioning within broader causal literature.
The paper would benefit from a clearer discussion of how “summary causal graphs” relate to other frameworks for representing feedback and temporal structure—such as dynamic SCMs or equilibrium SEMs—to help readers integrate this work into the larger causal-modeling landscape.
6.	Conclusion and synthesis.
The conclusion appropriately reiterates the main contribution—extending identification of micro-level direct effects to cyclic and confounded systems—and connects it to applied domains such as epidemiology. However, after a dense theoretical development, it functions more as a summary than a synthesis. The paper would be stronger if the authors distilled the intuitive meaning of the main theorems and clarified concretely what these new identification results enable beyond prior frameworks (e.g., Avin et al., Pearl). The “future work” paragraph, while interesting, somewhat dilutes the central message by listing multiple directions rather than emphasizing the principal conceptual advance.

**Questions:**

1. Clarification of intuition. The definitions of the average controlled and average natural micro direct effects are mathematically precise but conceptually demanding. Could the authors expand the intuitive explanation—perhaps by adding a small time-indexed causal diagram or a simple dynamic example—to illustrate what these effects represent in a concrete feedback system?
2. Relationship to classical identification results. The natural micro direct effect appears to generalize nested counterfactual identification results from Pearl (2001) and Avin et al. (2005). Could the authors clarify in what sense their graphical conditions differ from, or strictly extend, those earlier frameworks? For example, do their results reduce to the known back-door or mediation-formula cases when the system is acyclic?
3. Interpretation of summary causal graphs. How should readers interpret the “summary causal graph” formally relative to dynamic structural causal models (DSCMs) or equilibrium SEMs? Is an SCG a representation of lag-specific causal relationships, or does it correspond to an equilibrium approximation over micro time scales?
4. Practical identifiability. The paper’s identifiability conditions seem strong. Could the authors comment on their practical plausibility—for instance, in what classes of dynamic systems (linear Gaussian, continuous-time diffusion, etc.) do these conditions typically hold?
5. Potential empirical validation. Even a small simulation illustrating how the proposed identifiability results could guide estimation would greatly help readers. Do the authors plan to test these ideas empirically, or are there conceptual obstacles that currently prevent such a demonstration?
6. Notation and constants. Theorem 3 introduces constants such as λ and r(M_{\text{micro}}). Could the authors briefly explain their interpretation or role in identifiability, and whether they have any empirical meaning?

---

> ### Author Response · Authors · 2025-11-18
>
> We thank the reviewer for this very thorough and insightful review. We will address each point raised one by one.
>
> 1. This is an interesting feedback, we will add a textual explanation of the difference between the CDE and the NDE to help the readers gain some intuition about the definitions.
>
> 2. This work is exclusively theoretical. While we do give some example of potential applications we leave real-world applications to future work.
>
> 3. There is no prior work that tackles the issue in a similar setting (CDE and NDE from SCG with hidden confounders without parametric assumptions). Therefore, it makes little sens to compare with other results. However, it is true that we did not cite Avin and Pearl,, we will cite it and discuss it in the final version.
>
> 4. For the main results (Theorems 1 and 2) we do not make any assumption regarding the SCG: the SCG can be cyclic and hidden confounders can be present. The only assumption (no hidden confounders) is used in the Proposition 1 for the sufficient conditions. The plausibility of this assumption highly depends on the application and has been thoroughly discussed in previous works. (in IT: "Root Cause Identification for Collective Anomalies in Time Series given an Acyclic Summary Causal Graph with Loops" by Assaad et al. in Public Health: "Data-driven model building for life-course epidemiology" by Petersen et al. and in Climate Science: "Explaining Satellite Anomalies–Causal Inference for Space Operations" by Schefels et al.)
>
> 5. Positioning within the existing literature is already done to some extent in the introduction but we are willing to do this more thoroughly in the final version. However, we would like to point out that to the best of our knowledge, there exists no other paper investigating the problem of identifying micro CDE and NDE using SCGs with hidden confounders.
>
> 6. This is interesting feedback, we will revise the conclusion to take into account this comment and provide more intuition in the conclusion.
>
> Questions:
> 1. We will add a textual explanation of the difference between CDE and NDE.
>
> 2. We will discuss a bit more how our results compare to existing results on micro effects in summary causal graphs. However, we would like to point out that to the best of our knowledge, there exists no other paper investigating the problem of identifying micro CDE and NDE using SCGs with hidden confounders.
>
> 3. The summary causal graph does not represent an equilibrium SEM. It represents qualitative relationships between variables when the lags are omitted.
>
> 4. Our results do not require any parametric assumptions. For a complete result in the linear case without hidden confounders, the reviewer can refer to "Identifiability of direct effects from summary causal graphs" Ferreira and Assaad.
>
> 5. If the reviewer finds it to be necessary we can add a simulation in appendix to show the validity of our results.
>
> 6. We are not sure what the reviewer is referring to. There is no Theorem 3, no $\lambda$ and no $r(M_{\text{micro}})$ in this paper.

---

### Official Review · Reviewer_uf7U · 2025-10-31

**Soundness:** 3
**Presentation:** 3
**Contribution:** 3
**Rating:** 4
**Confidence:** 4

**Summary:**

The authors investigate identifiability of direct effects from so-called Summary Causal Graphs (SCGs), which encode in a compact way full causal graphs often used in dynamical or temporal systems. More specifically, they investigate the identifiability of two types of direct effects: average controlled and average natural direct effects. The main contribution of the paper are new sufficient conditions (expressed in a graphical language) for identifying average controlled *micro* direct effects and average *natural micro* direct effect in SCGs in general non-parametric setting and in the presence of hidden confounding.

**Strengths:**

This article falls within the research area of ​​identifying causal effects using partially defined graph models as, e.g., CPDAGs and PAGs. Studies in this direction are very well motivated and important from both a theoretical and practical perspective. The authors consider summary causal graphs (SCGs), which are well known partially defined models that ignore temporal information and allow cycles. A key contribution of the work are new graphical identification criteria which extend the criteria proposed by Ferreira & Assaad (2024a) for SCGs but without hidden confounders and assuming only linear mechanisms. The proofs are based directly on Pearl's do calculus.

**Weaknesses:**

The main weakness is that the graphical identifiability criteria are not complete. Theorem 1, which is the main result of the paper, only gives sufficient conditions for identifiability. The question arises as to how large the difference is between identifiable cases and those that do not meet the conditions specified in the theorem. The authors note that these conditions are generally not necessary, but do not analyze this in more detail.

This issue is addressed in some way in Proposition 1 that says that the conditions do become necessary but only considering identifiability by adjustment and, what is important, under Assumption 1 which means that there are no hidden confounding in the model. But allowing hidden confounders is a key property considered in the paper.

Similar questions apply to the conditions in Theorem 2.

Another problem is that the authors do not discuss the algorithmic (and computational complexity) aspects of the graphical conditions (in Theorem 1 and 2). In particular, how can one compute the set of possible parents of PP(Y_t)?

**Questions:**

Please refer to my comments above. In particular, do you know instances violating the graphical conditions in Theorem 1 that are identifiable (with hidden confounders and not necessarily by the adjustment).

L. 151: ∀Y_t, ∀X_{t−γ} ∈ V \ {X_{t−γ}} should be: ∀Y_t, ∀X_{t−γ} ∈ V \ {Y_t}

Please give labels (a), (b), etc. in Figures, as e.g. in Fig.1.

---

> ### Author Response · Authors · 2025-11-18
>
> We thank the reviewer for the relevant comments. We will address each issue.
>
> * The reviewer is correct to point out the absence of necessary conditions. While Proposition 1 gives some insights on under which assumptions is the criterion of Theorem 1 necessary, we do not provide intuition to find a general complete result. However, our results are sufficiently general to be applied in many real world examples as is shown in Figure 6 and in our comment to reviewer HU17. Moreover, we would like to emphasize that finding necessary and sufficient conditions for identifiability of micro effects in summary causal graphs with hidden confounders is a very hard problem as all previous work has pointed out. The combination of 1) hidden confounders, 2) the abstraction of the graph, and 3) micro queries, complicates the identifiability process.
>
> * This comment is very interesting, we thank the reviewer for this idea and we will make sure to include it in the final version. Computing the algorithmic complexity of our results is a fairly easy problem. As Theorem 1 only requires to find the strongly connected component of $Y$ and the ancestors of $Y$, its complexity is linear in the number of vertices. Theorem 2 requires to find the strongly connected components of $X$ and $Y$ and the ancestors of $Y$ which can be done in linear time. Moreover, it requires to find the possible parents of $X_{t-\gamma}$ and $Y_t$ which can be directly computed using their respective parents and $\gamma_{max}$ as is described in Definition 6. Therefore, Theorem 2 has complexity in $O(|V|*\gamma_{max})$.
>
> * Thank you for pointing out the typos, this will be fixed in the final version.

---

> > ### Comment · Reviewer_uf7U · 2025-11-28
> > **Further questions**
> >
> > Dear Authors, thank you very much for your responses and comments. Below, my further concerns regarding your paper.
> >
> > I agree that finding  sound and complete criteria is a very challenging task. Since your results seem to propose the first criteria for identifications of direct effects from SCGs in a non-parametric setting, it is important to explain to what extent your criteria "outperform" a naive approach. What I mean here is that, e.g., the assumption in Theorem 1 that Scc(Y,G^s) ⊆ {Y}, i.e., there is no cycle in G^s that includes Y or the only cycle that includes Y is a self loop on Y, seems to be quite strong (and not very surprising).
> >
> > Regarding the issue of computational complexity: I may have misunderstood your definition of possible parents (Definition 6). But the definition is indeed unclear. You say: The set of possible parents of the temporal vertex $Y_t$ is the set of temporal vertices (i.e. vertices in compatible FT-ADMGs) which are parents of $Y_t$ in at least one compatible FT-ADMG.
> >
> > According to this description, in Fig. 3 $X_{t-1},W_{t-1}$ is a set of possible parents of $W_t$ since the nodes are parents of $W_t$ in the compatible FT-ADMG (c). But also $X_t, X_{t-1},W_{t-1}$ are possible parents of $W_t$ since the nodes are parents of $W_t$ in the compatible FT-ADMG (b). So it seems there may be several (different) sets of possible parents. However, it seems that you understand the set of possible parents just as the set of parents of $Y$ in SCG (with additional indices $t-\gamma$). Could you please clarify this?
> >
> > Further questions:
> >
> > "Therefore, Theorem 2 has complexity in $(|V|* \gamma_{max})$." You mean: in $(|E| + |V|* \gamma_{max})$?
> >
> > In the Paper in Property 1: what do you mean by "$do(Parents(Y_t,G) = z |_{Parents(Y_t,G))}"

---

> > > ### Author Response · Authors · 2025-11-28
> > >
> > > Thank you for taking the time to help us improve our paper.
> > >
> > > Regarding the first point. While our results are not complete we do not believe that they  are not trivial. Indeed the condition $Scc(Y,G^s) \\subseteq \\{Y\\}$ seems to us very necessary.
> > >
> > > Take for example the SCG $G^s_1 = (\\{X,Y\\}, \\{(X,Y),(Y,X)\\})$. In this graph $Scc(Y,G^s_1) = \\{X,Y\\}$ and the CDE of $X$ on $Y$ is not identifiable. To see this consider the FT-ADMGs $G_{1,1} = (\\{X_{t-1}, X_{t}, Y_{t-1}, Y_{t}\\},\\{(X_{t-1},Y_t),(Y_{t-1},X_t),(X_{t-1},Y_{t-1}),(X_t,Y_t)\\})$  and $G_{1,2} = (\\{X_{t-1}, X_{t}, Y_{t-1}, Y_{t}\\},\\{(X_{t-1},Y_t),(Y_{t-1},X_t),(Y_{t-1},X_{t-1}),(Y_t,X_t)\\})$. Both $G_{1,1}$ and $G_{1,2}$ are compatible with $G^s_1$ but in $G_{1,1}$ one has $P(Y_t\\mid do(Parents(Y_t,G_{1,1}))) = P(Y_t\\mid X_{t-1}, X_t)$ whereas in $G_{1,2}$ one has $P(Y_t\\mid do(Parents(Y_t,G_{1,2}))) = P(Y_t\\mid X_{t-1})$.
> > >
> > > The intuition is that if there is a cycle on $Y$, then there exists at least one variable $X$ which is a parent of $Y$ and a descendant of $Y$. Because of this, there exists two compatible FT-ADMG, one in which $X_{t}$ is a parent of $Y_t$ and one in which $X_t$ is a descendant of $Y_t$. This duality of the variable $X_t$ makes it impossible to identify the CDE.
> > >
> > > Now regarding the second point on complexity, we have agreed with reviewer aDLB to clarify definition 6 (Possible Parents). Here is the new version:
> > > $$PP(Y_t) = \\{ P_{t-\\gamma} \\mid P \\in Parents(Y,G^s), \\gamma \\in [0,\\gamma_{max}]\\} \\backslash \\{Y_t\\}$$
> > > This set of "possible parents" is exactly the set of temporal variables which are parents of $Y_t$ in at least one FT-ADMG compatible with $G^s$.
> > > To compute this set, one could compute all the compatible FT-ADMGs $\\{G_i \\mid G_i \\text{compatible with } G^s\\}$ and then compute for each $G_i$ the set of parents of $Y_t$ and then compute the union of all those sets (ie, $PP(Y_t) = \\bigcup_{G_i} Parents(Y_t,G_i)$).
> > > **However, this method is overly expensive and not necessary as the first definition provided gives us a much more efficient algorithm: one just needs to compute the set of parents of $Y$ in the SCG $G^s$ the then take every past instants of these variables up to $t-\\gamma_{max}$. This allows us to compute the possible parents in $O(|V|\*\\gamma_{max})$**.
> > >
> > > To illustrate this definition, take the example of Figure 3. If one is looking for the possible parents of $W_t$ then one has two choices.
> > > Firstly, one can compute every compatible FT-ADMG (in this case we can limit ourselves to the 2 FT-ADMGS given in Figure 3.b and 3.c) and compute for each FT-ADMG the set of parents of $W_t$ and then take the union of these sets of parents. In this example, one would obtain $Parents(W_t) = \\{X_{t-1},W_{t-1},X_t\\}$ in the FT-ADMG of Figure 3.b and $Parents(W_t) = \\{X_{t-1},W_{t-1}\\}$ in the FT-ADMG of Figure 3.c. Thus after taking the union of these two sets, one will find that $PP(W_t) = \\{X_{t-1},W_{t-1},X_t\\}$.
> > > Secondly, one can use the definition provided and compute directly the set of parents of $W$ in the SCG of Figure 3.a and then take the past instants of these variables without having to compute every compatible FT-ADMG. Using this, one obtains $Parents(W) = \\{W,X\\}$ and thus $PP(W_t) = \\{W_{t-1},W_t,X_{t-1},X_t\\}\\backslash\\{W_t\\} = \\{W_{t-1},X_{t-1},X_t\\}$.
> > > Of course, both methods output the same answer, but the second one is much faster as there is no need to enumerate all compatible FT-ADMG.
> > >
> > > The last paragraphs show that finding the possible parents of a node has complexity $O(|V|\*\\gamma_{max})$ and a set of possible parents is of size at most $|V|\*\\gamma_{max}$. Moreover, finding the set of ancestors of a node has complexity $O(|V|)$ and finding the strongly connected component of a node has complexity $O(|V|)$. Lastly, computing the intersection of 2 sets has complexity $O(n)$ where $n$ is the size of the sets. Notice that Theorem 2 requires finding the strongly connected components of 2 nodes, finding the ancestor sets of 2 nodes and finding the possible parents of 2 nodes and taking their intersection. Therefore, Theorem 2 has complexity $2\*O(|V|) + 2\*O(|V|) + 2\*O(|V|\*\\gamma_{max}) + O(|V|\*\\gamma_{max}) = O(|V|\*\\gamma_{max})$.

---

> > > > ### Author Response · Authors · 2025-11-28
> > > >
> > > > Regarding your last question about property one, by $\\mathbb{z}\\mid_{Parents(Y_t,G)}$ we meant the subset of $\\mathbb{z}$ that corresponds to variables in $Parents(Y_t,G)$. However, we agreed with reviewer aDLB that the notation $\\mathbb{z}\\mid_{Parents(Y_t,G)}$ was unclear and that we will modify it. In the new version we will replace the notation $\\mathbb{z}\\mid_{Parents(Y_t,G)}$ with $\\mathbb{p}$ which represents a possible value of $Parents(Y_t,G)$, we will also add a clarifying sentence stating that $\\mathbb{p}$ corresponds to the subset of values in $\\mathbb{z}$ associated with $Parents(Y_t,G)$.
> > > >
> > > > To clarify further what we mean: $\\mathbb{Z} = PP(Y_t)$, $\\mathbb{P} = Parents(Y_t,G)$, $\\mathbb{z}$ is a set of possible values for the set of variables $\\mathbb{Z}$ and $\\mathbb{p}$ is a set of possible values for the set of variables $\\mathbb{P}$. We know that $Parents(Y_t,G) \\subseteq  PP(Y_t)$ (ie, $\\mathbb{P} \\subseteq \\mathbb{Z}$) so we can force $\\mathbb{p}$ and $\\mathbb{z}$ to match, that is to say for every variable that is in both in $\\mathbb{Z}$ and in $\\mathbb{P}$, the possible value corresponding to this variable is the same in $\\mathbb{z}$ and in $\\mathbb{p}$.
> > > >
> > > > Thank you again for the interesting questions and valuable feedback.

---

### Official Review · Reviewer_HU17 · 2025-10-31

**Soundness:** 3
**Presentation:** 2
**Contribution:** 3
**Rating:** 4
**Confidence:** 2

**Summary:**

This paper investigates the identifiability of micro direct effects within summary causal graphs (SCGs), which are abstractions used to represent complex dynamic causal systems. This work extends previous research on the identification of micro direct effects in summary causal graphs, which primarily relied on linear assumptions, by instead considering the more realistic non-linear case to identify non-parametric direct effects. The paper's main contributions are providing sufficient conditions to identify the controlled micro direct effect (CDE) and the natural micro direct effect (NDE) from SCGs in the presence of hidden confounding. Furthermore, it shows that these conditions for CDE become necessary in the absence of hidden confounding when identification is restricted to adjustment.

**Strengths:**

- The paper tackles the identification of micro direct effects in non-parametric settings, an open challenge in summary causal graphs for causal inference.
- The paper establishes sufficient graphical conditions for identifying both the average controlled micro direct effect and the average natural micro direct effect.
- The paper effectively uses graphical examples (e.g., Figs 1-4) to demonstrate why identification is difficult, showing how different underlying time-series graphs can be compatible with the same SCG.

**Weaknesses:**

- The paper does not establish necessary and sufficient conditions for CDE in the general case with hidden confounding or for identification methods beyond simple adjustment.
- The paper relies on the assumption of a known maximal lag between causes and effects without adequately justifying why this is a reasonable or practical assumption in real-world applications.
- The examples provided can create a pessimistic impression, as they frequently illustrate scenarios where the CDE and NDE are not identifiable, suggesting such cases may be common, even with the sufficient conditions provided in this paper.
- The paper's contributions are purely theoretical and lack empirical validation through numerical experiments or applications to real-world data.

**Questions:**

Compared to the technical difficulty of identifying micro total effects (for which conditions in a non-parametric setting for SCGs have been established in previous work ), what are the specific technical challenges involved in identifying micro direct effects, as addressed in this paper?

---

> ### Author Response · Authors · 2025-11-17
>
> We thank the reviewer for this insightful question. We will address  points raised  one by one.
>
> * We fully acknowledge that our paper does not provide a complete identification theory for the CDE beyond adjustment. Our goal was rather to lay the foundations for identifying micro CDEs within SCGs and to show that, under sufficient graphical conditions, adjustment provides a sound and interpretable solution. We also wish to emphasize that, in the applied causal inference literature, adjustment remains by far the most commonly used and interpretable identification strategy for direct and total effects. Focusing on this setting therefore allows our results to connect directly with standard applied practices.
>
> * In this work, we assume stationarity. Without this assumption, the estimation of a direct effect from data would be ill-defined, since we consider a single multivariate time series in which each temporal variable has only one observed value. Given stationarity, it is reasonable, and in fact natural, to assume the existence of a maximal lag between a cause and its effect. In fact, rather than being an additional modeling assumption, the existence of a maximal lag is a direct consequence of the stationarity assumption. We will clarify this point in the revised version of the paper.
>
> * While we recognize that the conditions of identifiability can be strong, we would like to emphasize that the CDE shown in figure 6 is in fact identifiable. Moreover, we will add in the final version an example of real world application in IT monitoring taken from "Identifiability of total effects from abstractions of time series causal graph" by Charles K. Assaad, Emilie Devijver, Eric Gaussier, Gregor Goessler and Anouar Meynaou Figure 6.1.c in which both the CDE and the NDE are identifiable. These examples illustrate the usefulness of our results.
>
> * The existing results on identifying micro total effects in SCGs cannot be directly applied to the identification of the CDE or the NDE. One key reason is that prior work focuses on total effects of a single treatment variable on an outcome, whereas in our setting, the CDE corresponds to the total effect of a set of treatments (namely, the parents or potential parents of $Y_t$) on $Y_t$. To the best of our knowledge, the existing literature does not provide identification results for total effects of multivariate treatments directly from the SCG (without enumerating the corresponding DAGs). Therefore, although our contribution is centered on identifying micro-level direct effects rather than total effects, we hope that our formulation will motivate and guide future research toward establishing general identification results for total effects of multivariate treatments within the SCG framework. Furthermore, we emphasize that unlike most paper in the literature focusing on identifying total effects by adjustment, in this work we take into account hidden confounders which adds a level of complexity.

---

### Official Review · Reviewer_aDLB · 2025-11-02

**Soundness:** 3
**Presentation:** 3
**Contribution:** 3
**Rating:** 8
**Confidence:** 3

**Summary:**

The paper studies the identification of controlled direct effects (CDEs) and natural direct effects (NDEs) [Pearl, 2001] in summary causal graphs, which are abstractions of time-series causal graphs and hence may contain cycles and hidden confounders. In particular, the paper proposes sufficient conditions under which the CDEs and NDEs are identifiable, and necessary conditions for identifying CDEs when there is no latent confounder.

**Strengths:**

- In general, the paper is well structured and clearly written. The definitions and theorems are formulated clearly, and detailed proofs are provided in the appendix.
- The subject of identifying direct effects is a good starting point, which can potentially lead to more general results on causal effects.
- The paper includes sufficient background on the definitions of two different direct effects, the time-series causal graphs, and identification methods such as do-calculus, which facilitates the understanding of the main results. I also found the illustrative examples and the real-world example (Section 6) very helpful.

**Weaknesses:**

1. While the paper provides sufficient conditions for identifying direct causes, it currently does not present necessary conditions in more general settings beyond identification by adjustment. As the authors also mentioned in the conclusion section, this is a limitation and could be a subject of future work.
2. Line 158: "a stationarity assumption becomes necessary to satisfy the positivity assumption." Could you elaborate on this? These two concepts seem quite distinct to me.
3. I believe it would be helpful for general readers if the authors could provide a concrete example illustrating the difference between controlled and natural directed effects near their definitions.
4. Line 283-286 (Property 1): The notation $z|_{Parents(Y_t,G)}$ is undefined.
5. Line 305 and below: The independence signs look tiny.
6. Line 460-461: Theorem 2 only shows a sufficient condition for identifiability, so it cannot be used to justify unidentifiability. Please provide more detailed explanations on why the NDE is unidentifiable.

**Questions:**

1. Is Assumption 1 identical to the causal sufficiency assumption commonly used in causal literature? If so, it may be worth explicitly mentioning this.
2. Definition 6. Is there a reason why the self-loop is stated explicitly? Why not treat $Y$ as a parent of itself when there is a self-loop?
3. Line 418 (Theorem 2): Does the identifying formula still qualify as an adjustment formula? Please clarify if it does not.

---

> ### Author Response · Authors · 2025-11-17
>
> We thank the reviewer for the constructive feedback and valuable comments. We will address several points raised then we will reply to each question one by one.
>
> About the stationarity and positivity assumptions:
> Since we are working with a single (non-multiple) multivariate time series, for each temporal variable $X_t$ we have only one observed value.
> Consider the temporal variables $X_t$ whose only observed value is $x_0$.
> Suppose that the domain of $X_t$ contains more than one possible value (i.e., more than just $x_0$) which is a natural assumption, otherwise this random variable would be constant.
>
> In this case, the positivity assumption is not satisfied because in the observed data we have $\Pr(X_t = x_1) = 0$ for all  $x_1 \neq x_0$. Hence, positivity cannot hold in this single-instance setting.
> Stationarity along with the maximal lag assumption allows us to consider each time instant of a temporal value as an observed value of this random variable and thus renders positivity possible.
> Take the following example: $X_{t-1} = x_1, X_{t} = x_0, Z_{t-1} = z_1, Z_{t}= z_0$ then without the stationarity assumption, we have $\Pr(X_{t} = x_1, Z_{t} = z_1) = 0$. However, with the stationarity assumption, we have $\Pr(X_{t} = x_1, Z_{t} = z_1) = 0.5$.
>
>
> About Property 1: We will replace the expression $z \mid_{Parents(Y_t, G)}$ by the simpler notation $\mathbb{p}$, which represents a possible value of $Parents(Y_t, G)$.
> We will also add a clarifying sentence stating that $\mathbb{p}$ corresponds to the subset of values in $\mathbb{z}$ associated with $Parents(Y_t, G)$.
>
>
> About the real-world example: In the current version, we intended to show an example where the NDE is not identifiable \textbf{using Theorem 2} (and not necessarily non-identifiable in general).
> We agree that the current formulation may be unclear, so we will revise it as follows:
> "The micro CDE can be identified using Theorem 1. However, the NDE cannot be identified using Theorem 2. We emphasize that this does not imply that the NDE is not identifiable in general."
>
> In the following we give a response to each question:
>
> 1- Yes Assumption 1 is identical to the causal sufficiency assumption. We will clarify this in the text.
>
> 2- The main reason for separating the self-loop of $Y$ from its other parents is that, for parent variables different from $Y$,  $\gamma$ may be equal to zero.
> However, when $Y$ treating $Y$ as a parent, $\gamma$ cannot be zero.
> We agree that our initial formulation is overly complex and can be simplified as suggested, for instance by adding a backslash $Y_t$ to explicitly exclude the case where $\gamma = 0$ for $Y$.
>
>
> 3- By adjustment, we refer to cases where the probability $P(y_t \mid do(x_{t-\gamma}))$ used in the expectation can be expressed as $P(y_t \mid do(x_{t-\gamma})) = \sum_{z} P(y_t \mid x_{t-\gamma}, z) P(z)$.  Given this definition, the identifying expression presented in Theorem 2 does not qualify as an adjustment formula. We will clarify our use of the term adjustment in the preliminaries section of the revised manuscript.

---

### Meta-Review · Area_Chair_62Yo · 2025-12-22

**Summary:**

The most problematic points all have to do with a  lack of mathematical rigor:
- Incompleteness of  graphical identifiability criteria,
- lack of necessary conditions for interpretability (Theorem 1),
- potential inconsistencies: Proposition 1 relies on Assumption 1 (no hidden confounding). But allowing hidden confounders  seems to be a key property considered in the paper.
- missing algorithmic (and computational complexity) aspects of the graphical conditions (in Theorem 1 and 2),
- Theorem 2 only shows a sufficient condition for identifiability, but it is used to justify unidentifiability,
- the main arguments rely on the assumptions which are not adequately justified, such as  assuming a maximal lag between causes and effects)

**Reviewer Concerns:**

Honestly, I don't think that any of these more fundamental points of criticism have been convincingly addressed in the rebuttal. I acknowledge that the rebuttal includes a new version of definition 6 (which was unclear in the paper), but it seems to me that this novel definition is a substantial change that would lead to several follow-up changes. So I think that a major revision would be be needed, and that the paper is currently not ready for publication.

**Reviewer Scores:**

I don't think that the rebuttal has led to significant changes in any of the scores.

---

### Decision · Program_Chairs · 2026-01-26

Reject